# Pregnant women's daily patterns of well-being before and during the COVID-19 pandemic in Finland: Longitudinal monitoring through smartwatch technology

Hannakaisa Niela-Vilén[1]*, Jennifer Auxier[1], Eeva Ekholm[2], Fatemeh Sarhaddi[3], Milad Asgari Mehrabadi[4], Aysan Mahmoudzadeh[3], Iman Azimi[3], Pasi Liljeberg[3], Amir M. Rahmani[4,5,6], Anna Axelin[1,2]

1 Department of Nursing Science, University of Turku, Turku, Finland, 2 Department of Obstetrics and Gynaecology, Turku University Hospital and Faculty of Medicine, University of Turku, Turku, Finland, 3 Department of Future Technologies, University of Turku, Turku, Finland, 4 Department of Electrical Engineering and Computer Science, University of California Irvine, Irvine, California, United States of America, 5 Department of Computer Science, University of California Irvine, Irvine, California, United States of America, 6 School of Nursing, University of California Irvine, Irvine, California, United States of America

⊕ These authors contributed equally to this work.
* hmniel@utu.fi

## Abstract

### Background

Technology enables the continuous monitoring of personal health parameter data during pregnancy regardless of the disruption of normal daily life patterns. Our research group has established a project investigating the usefulness of an Internet of Things–based system and smartwatch technology for monitoring women during pregnancy to explore variations in stress, physical activity and sleep. The aim of this study was to examine daily patterns of well-being in pregnant women before and during the national stay-at-home restrictions related to the COVID-19 pandemic in Finland.

### Methods

A longitudinal cohort study design was used to monitor pregnant women in their everyday settings. Two cohorts of pregnant women were recruited. In the first wave in January-December 2019, pregnant women with histories of preterm births (gestational weeks 22–36) or late miscarriages (gestational weeks 12–21); and in the second wave between October 2019 and March 2020, pregnant women with histories of full-term births (gestational weeks 37–42) and no pregnancy losses were recruited. The final sample size for this study was 38 pregnant women. The participants continuously used the Samsung Gear Sport smartwatch and their heart rate variability, and physical activity and sleep data were collected. Subjective stress, activity and sleep reports were collected using a smartphone application developed for this study. Data between February 12 to April 8, 2020 were included to cover four-week periods before and during the national stay-at-home

**Data Availability Statement:** The COVID-19 and well-being of pregnant women study data include

sensitive health information, and the informed consent signed by the participants does not allow the data to be made publicly available due to ethical restriction. According to the current approval by the Ethics Committee of the Hospital District of Southwest Finland, the participants gave permission to use the collected data only for the purpose described in the consent. Data requests may be subject to individual consent and/or ethics committee approval. Researchers wishing to use the data should contact the Ethics Committee of the Hospital District of Southwest Finland, contact details: Tyks U-hospital, Kiinamyllynkatu 4-8, UB3, PO Box 52, FI-20521 TURKU, Finland, e-mail: eettinen.toimikunta@tyks.fi. We recommend first to contact the PI of the research project, associate professor Anna Axelin, contact details: University of Turku, Department of Nursing Science, 20014 University of Turku, Finland, e-mail: anmaax@utu.fi.

**Funding:** This research was supported by the Academy of Finland https://www.aka.fi/en/ (Awards 313448 (AMR), 313449 (AA), 316810 (AMR), and 316811 (AA)) and U.S. National Science Foundation https://www.nsf.gov/ (Award CNS-1831918) (AMR). The funders had no role in study design, data collection and analysis, decision to publish, or preparartion of the manuscript.

**Competing interests:** The authors have declared that no competing interests exist.

restrictions. Hierarchical linear mixed models were exploited to analyze the trends in the outcome variables.

## Results

The pandemic-related restrictions were associated with changes in heart rate variability: the standard deviation of all normal inter-beat intervals (p = 0.034), low-frequency power (p = 0.040) and the low-frequency/high-frequency ratio (p = 0.013) increased compared with the weeks before the restrictions. Women's subjectively evaluated stress levels also increased significantly. Physical activity decreased when the restrictions were set and as pregnancy proceeded. The total sleep time also decreased as pregnancy proceeded, but pandemic-related restrictions were not associated with sleep. Daily rhythms changed in that the participants overall started to sleep later and woke up later.

## Conclusions

The findings showed that Finnish pregnant women coped well with the pandemic-related restrictions and lockdown environment in terms of stress, physical activity and sleep.

## Introduction

The disruption of societal patterns results in negative effects for pregnant women, but the extent and importance of such negative effects are unclear. Pregnancy and newborn outcomes that have been monitored after natural or manmade disasters are inconsistent [1]. Some negative impacts were noted for pregnant women and their newborns after the 2008 global economic crisis and the World Trade Center attacks in 2001. However, many of these effects were observed through single-time-point assessments of pregnant women's stress levels, or the studies retrospectively analysed pregnancy and newborn health outcomes years after the events had occurred [1, 2]. Psychological distress as a response to traumatic events has been reported, but little work has been done to investigate daily patterns of wellness during disasters [1]. Thus, post-disaster pregnancy surveillance should be improved.

Identifying increased levels of stress and disruptions in daily patterns of pregnant women's physical activity and sleep is important, as they may be deemed indicators for well-being in pregnancy [3]. High levels of maternal antenatal stress are associated with an increased risk of pregnancy complications, such as preterm birth [4]. Often, physical activity and its intensity decrease as pregnancy progresses. Many women are inactive during pregnancy despite the evidence that moderate, low-risk activities are safe and beneficial during all stages of pregnancy [5]. Sleep disturbances, such as insomnia and sleep fragmentation stemming from hormonal and physiological changes, are common during pregnancy [6, 7]. Stress, physical activity and sleep are interdependent; for example, stress may escalate sleep disorders, which have harmful effects on maternal health [8]. Physical activity in pregnancy has been linked to sustained mobility, healthy sleep patterns and usual life activities [9]. Adequate physical activity and sleep can also mitigate high levels of stress during pregnancy [10].

Modern technology enables the continuous monitoring, tracking and transmitting of personal health parameter data during pregnancy [11]. In light of the current disruption of normal daily life patterns due to the COVID-19 pandemic, the use of wearable devices and mobile apps for collecting personal health parameters for use by health care providers and public

health agencies is highlighted among many health care organizations and areas of clinical practice. The continuous monitoring of maternal antenatal stress, physical activity and sleep parameters before and during the current COVID-19 could provide unique information about any disruptions in the well-being of pregnant women. The aim of this study was to examine daily patterns of well-being (stress, physical activity and sleep) in pregnant women before and during the national stay-at-home restrictions related to the COVID-19 pandemic in Finland.

## Methods

### Study design

A longitudinal cohort study design was used. This study is part of a project investigating the usefulness of an Internet of Things (IoT)-based system and smartwatch technology for monitoring women during pregnancy to explore variations in stress, physical activity and sleep. The ultimate goal of the project is to develop a ubiquitous monitoring and early detection and prevention system for pregnant women. The interest is in variations in stress defined by heart rate variability, physical activity indicated with step counts and duration of sleep; all the parameters are followed using a smartwatch. The participants were asked to wear a smartwatch continuously from early pregnancy (gestational week 12–15) until three months postpartum. In our pilot work, long-term monitoring with a smart wristband was evaluated feasible among pregnant women [11, 12]. A cohort within the sample from the ongoing study had specific exposure to current events in Spring 2020 related to regulations for preventing the spread of the SARS-CoV-2 (i.e. stay-at-home orders and travel bans).

### Study participants

The eligible participants were Finnish-speaking women with singleton pregnancies at gestational weeks 12–15. They each had to have a smartphone with Android or iOS as an operating system. The participants were recruited through social media advertisements or maternity clinics in two waves: 1) pregnant women with histories of preterm births (gestational weeks 22–36) or late miscarriages (gestational weeks 12–21) were recruited in January-December 2019, and 2) pregnant women with histories of full-term births (gestational weeks 37–42) and no pregnancy losses were recruited between October 2019 and March 2020.

Pregnant women contacted the researchers by email, and/or the researcher phoned the interested women, and based on their initial communication and verbal study information, a meeting was scheduled. A total of 62 pregnant women were recruited. At the time of the COVID-19 pandemic and restrictions in Finland, 21 women had given birth and therefore were excluded. Three women were withdrawn from the study. Thus, the final sample size for this study was 38 pregnant women, with eight of them belonging to the high-risk pregnancy group.

### Data collection

Each participant was provided a Samsung Gear Sport smartwatch, which has shown acceptable validity regarding sleep in everyday context [13] and regarding step count in a treadmill test [14]. The participants were asked to wear the smartwatches continuously from the recruitment until three months postpartum, and they were to send the data from the smartwatches daily or at least weekly. An IoT-based monitoring system was developed (Fig 1). The photoplethysmography (PPG) and inertial measurement unit (IMU) sensors of the smartwatch were utilized. A Tizen-based application was developed to collect 12 minutes of the signals—with a 20-Hz sampling frequency—every second hour. The setup was determined to acquire sufficient data

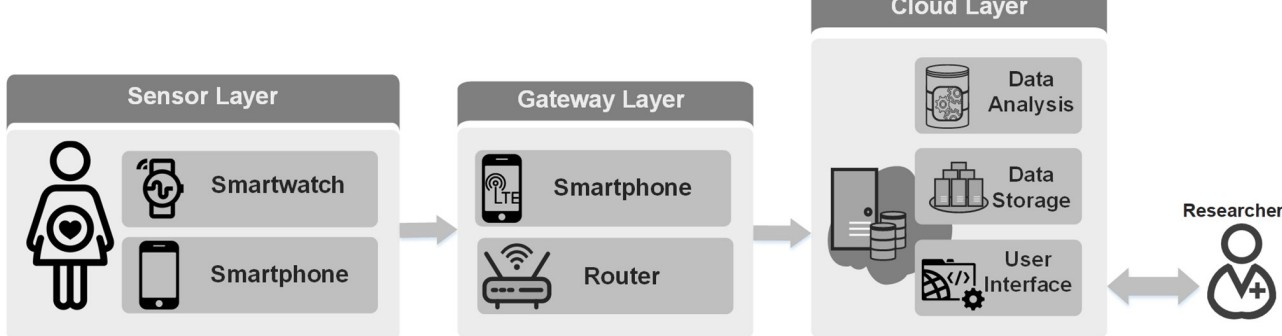

**Fig 1. IoT-based remote monitoring system.** First, the sensor layer performs health data collection, utilizing a smartwatch and a smartphone. Second, the gateway layer acts as a bridge between the devices and the remote server, transmitting the collected data to our remote servers. Third, the cloud layer is responsible for data storage and data analysis. A user interface is also provided for data visualization.

for heart rate and heart rate variability (HRV) analysis, considering the smartwatch's limited battery life. HRV can be used as an objective assessment of stress in pregnant women [15, 16]. However, interpreting HRV is challenging due to individual variations and many variables influencing the values. Long-term monitoring of participants enables to detect trends in HRV and possible novel data about pregnant women [17]. The smartwatch also provided continuously collected physical activity data (step counts and inactive time) and sleep events (sleep start and end times). Thus, the total sleep time (TST) and wake after sleep onset (WASO) could be determined.

A cross-platform mobile application was developed for the subjective data collection. The application was installed in the participants' smartphones. The participants used the application to complete the questionnaires and weekly questions. A background questionnaire and validated questionnaires concerning depression (EPDS) [18] and pregnancy anxiety (PRAQ-R2) [19] were completed after the recruitment. In weekly questions that appeared in the application every seven days, the participants assessed their levels of stress, levels of activity and sleep quality during the past week with a numerical scale from 0 to 100. Zero indicated no stress/ no physical activity/ very poor sleep quality, and 100 indicated a very high level of stress/ very high level of physical activity/ very good sleep quality. On March 18, a weekly question related to the COVID-19 pandemic was added to the application. The participants were asked to assess their levels of worry regarding the epidemic in Finland with a numerical scale of 0–100. The weekly questions were developed for this study and therefore not validated. The participants could also contact the researchers via the application, if necessary. However, two-way communication was not possible, and the researchers responded by email. Note that the participants could not view their data via the application.

Data between February 12 to April 8, 2020, were included to cover four-week periods before and after March 12, 2020, which was a significant date in Finland regarding the COVID-19 pandemic. In general, most Finnish people acknowledged the seriousness of the SARS-CoV-2 virus and COVID-19 disease. On March 12, the Finnish government recommended canceling all large public events and recommended avoiding close contact with other people, especially with at-risk groups. Since that day, national restrictions tightened quite rapidly. On March 16, a nationwide lockdown was declared, and people were strongly urged to stay at home, maintain social distancing and follow careful hygiene procedures. Schools and universities, as well as libraries, museums and other public venues, were closed, and working at home was strongly

recommended. Outdoor activities alone or with adequate distance from others have been allowed throughout the restrictions.

## Data preprocessing

**Stress.** We exploited PPG signals to extract heart rate and HRV parameters. PPG is a simple and energy-efficient optical method that is widely used to remotely track heart activities [17, 20]. However, the method is susceptible to motion artifacts and environmental noises that are inevitable in home-based health monitoring. Therefore, we developed a method for removing unreliable signals, thus preventing invalid analysis and misinterpretation. In this regard, we utilized morphological features of PPG signals to automatically label every 10 seconds of PPG signals to be able to select reliable signals for the analysis [21].

HRV analyses were performed with five-minute recordings of reliable PPG signals collected during sleep time when the resting heart rate level was the lowest and less artifacts existed [22]. The sampling frequency of the collected PPG signals was 20 Hz. Therefore, HRV parameters that were with low error rates using the 20-Hz PPG signals were extracted [23].

For HRV extraction, we first implemented a peak detection method for retrieving the peaks corresponding to each heartbeat. Second, normal inter-beat intervals (NNIs) were computed. Then, for every reliable five-minute signal, we calculated the root mean square of the successive differences (RMSSD) of NNIs, the standard deviation of all NNIs (SDNN), low-frequency (LF) power (0.04–0.15 Hz), high-frequency (HF) power (0.15–0.4 Hz) and the LF-to-HF ratio (LF/HF). The analysis was implemented using the HeartPy and SciPy libraries in Python [24]. The outliers were removed, and the average values of HRV parameters during sleep were obtained.

**Physical activity and sleep.** The participants' step counts, inactive time and sleep parameters were provided via the Samsung watch. The daily data concerning physical activity were considered to be valid if the user wore the device for at least 10 hours while awake. The wearing time was computed with the granularity of five minutes, utilizing logged events obtained through the watch, such as hand movements, step counts, the heart rate and sleep.

Two sleep parameters per night were calculated: TST and WASO. TST refers to the total time that our subject was sleeping during the night. WASO is the total awake time after the start of sleep and before the final awakening. To validate sleep intervals for such sleep events, we visualized various sources of data, including the steps and hand movements that the watch reported. Using such visualizations, we were able to retrieve actual sleep intervals by manually looking at the data.

## Statistical analysis

The characteristics of the participants were compared between the high-risk and low-risk groups with chi-square test, or two-sample t-test (Mann-Whitney U-test if non-normal data). Hierarchical linear mixed models were exploited to analyze the trends in between-person and within-person changes in the dependent variable of interest. We did this using the notation that Raudenbush-Bryk defined, as well as the recommendations of Bolger-Laurenceau [25, 26]. We considered daily values in two different time intervals: four weeks before and four weeks during the COVID-19 pandemic (a total of 56 days), including 38 subjects in each interval. All participants were combined in one group in the analyses since there were no significant differences in their background characteristics, except gestational weeks during the study (Table 1). The dependent variables in these models are measurements related to HRV (the SDNN, RMSSD, HF power, LF power), physical activity (step counts, inactive time) and sleep (TST, WASO). In the model, the single within-subject independent variable was time (day), and the between-subject independent binary variables were the pandemic group (before or

**Table 1. The characteristics of the participating pregnant women.**

| Variable | Participants n = 38 | With high-risk pregnancy n = 8 | With low-risk pregnancy n = 30 | p value |
|---|---|---|---|---|
| Age (years), *mean (SD)* | 31.4 (4.3) | 32.1 (3.6) | 31.2 (4.5) | 0.595 |
| Gestational weeks at March 12[th], *mean (SD)* | 23.5 (6.1) | 29.9 (1.9) | 21.7 (5.7) | <0.001 |
| BMI before pregnancy, *mean (SD)* | 26.3 (6.0) | 25.4 (4.0) | 26.7 (6.3) | 0.820 |
| Planned pregnancy, *n (%)* | 33 (87) | 6 (75) | 27 (90) | 0.265 |
| Chronic illness*, *n (%)* | 11 (29) | 4 (50) | 7 (23) | 0.139 |
| Smoking during pregnancy, *n (%)* | 0 (0) | 0 | 0 | - |
| Education, *n (%)* | | | | |
| Highschool | 18 (48) | 2 (25) | 16 (53) | 0.198 |
| Vocational | 10 (26) | 4 (50) | 6 (20) | |
| University | 10 (26) | 2 (25) | 8 (27) | |
| Occupation, *n (%)* | | | | |
| Paid work | 29 (76) | 5 (63) | 24 (80) | 0.233 |
| Unemployed | 1 (3) | 0 | 1 (3) | |
| Student | 5 (13) | 1 (12) | 4 (14) | |
| Other | 3 (8) | 2 (25) | 1 (3) | |
| Living with partner, *n (%)* | 37 (98) | 7 (88) | 30 (100) | 0.211 |
| Wearing smartwatch at work, *n (%)* | 35 (92) | 8 (100) | 27 (90) | 0.351 |
| Depressive symptoms (EPDS score[i]), *mean (SD)* | 5.0 (3.9) | 5.4 (3.0) | 4.8 (4.2) | 0.388 |
| Pregnancy-related anxiety (PRAQ-R2 score[ii]), *mean (SD)* | 38.2 (7.3) | 36.8 (5.4) | 38.6 (7.7) | 0.529 |

*migraine, colitis ulcerosa, hypothyroidism, fibromyalgia, SLE, or skin disease

[i]Scores from 0–30 *(High scores indicate risk for Depressive symptoms)*

[ii]Scores from 10–50 *(High scores indicate signs of pregnancy related anxiety).*

during the pandemic), and education level (university level or lower level education). Further, the study group (high-risk or low-risk pregnancy) was used as between-subject independent binary variable in the models to detect possible differences between the groups. Only night-time data were included in the analyses; thus, the time of the day was controlled. However, the complex variability of healthy heart could not be completely controlled in such a long-term monitoring with PPG signal [17, 22]. We used the Statsmodel package of Python to do the aforementioned analyses [27].

To deal with the outliers of the measurements in physical activity, sleep and heart rate variability, we re-scaled the values using z-score, and we kept entries with absolute values of less than 3 (it captures 99.73% of the data). Moreover, we leveraged the paired t-test to compare the difference in subjective data before and during the pandemic. The null hypothesis was a zero mean difference between the variables of interest. To visualize the difference between before and during the pandemic, we utilized the complementary cumulative distribution function (CCDF) and the Bland-Altman plot.

Missing data were handled using pairwise deletion by filtering data for all parameters classified as unreliable due to non-available data (i.e. errors in device recording and participants not wearing devices) as well as outlier recordings not classified as reliable [28].

## Ethical issues

A favourable statement (Dnro: 1/1801/2018) from the Ethics Committee of the Hospital District of Southwest Finland was obtained before data collection. A written informed consent was obtained from each participant.

## Results

### Participants

A total of 38 pregnant women participated in this study and were followed eight weeks from February 12 to April 8, 2020. Their gestational weeks during the COVID-19 outbreak in Finland were 23.5 (SD 6.1) weeks on average. At the recruitment, the participants had no depressive symptoms (mean EPDS score 5.0) or pregnancy-related anxiety (mean PRAQ-R2 score 38.2). There were no differences between the participants with high-risk and low-risk pregnancies except the gestational weeks during the study period, since the recruitment of women with high-risk pregnancies was performed earlier compared with the women with low-risk pregnancies (Table 1).

### Stress

The hierarchical linear models showed that pandemic-related restrictions were associated with changes in HRV. At the time the restrictions were set, the SDNN (p = 0.034), LF power (p = 0.040) and LF/HF ratio (p = 0.013) increased compared with the weeks before the restrictions. However, the pandemic-related restrictions over time were also associated with the SDNN (p = 0.008), showing a decreasing trend of the SDNN during the weeks during the restrictions. The pandemic-related restrictions showed no association with the RMSSD or HF power (Fig 2).

The participants' subjective levels of worry related to the pandemic in Finland were reasonably high. The mean value fluctuated from 61 (SD 21) via 56 (SD 18) to 60 (SD 20) during the three first weeks during the restrictions, respectively. Based on the t-test of the evaluations of the participants, their subjective stress levels were statistically significantly higher during the restrictions compared with the weeks before (p = 0.008) (Fig 3).

### Physical activity

Both pandemic-related restrictions (p = 0.001) and time (p = 0.013) were associated with the participants' daily total step counts. The step counts decreased as pregnancy proceeded and especially when the restrictions were set.

Correspondingly, the daily inactive time increased during the study period, as pandemic-related restrictions (p<0.001) and time (p<0.001) were both significantly associated with the inactive time when the restrictions were set. However, the pandemic-related restrictions over time were significantly associated with decreasing inactive time (p = 0.014) (Fig 4). The participants' subjective evaluations of their weekly physical activity did not change (Fig 3).

### Sleep

Pandemic-related restrictions were not associated with p = 0.266, the TST of the pregnant women. However, TST decreased (p = 0.021) as pregnancy proceeded. The pandemic-related restrictions were not significantly associated with the periods of WASO (p = 0.065).

The participants woke up a mean of 15 minutes (p = 0.007) later during the pandemic-related restrictions compared with the weeks before the virus outbreak. Correspondingly, they went to sleep approximately 10 minutes later; however, the change was not statistically significant (p = 0.0504). Subjectively evaluated quality of sleep did not change based on the participants' weekly evaluations (Fig 3).

## Discussion

The COVID-19 pandemic seemed not to have major impacts on the daily patterns of Finnish pregnant women. Their subjectively assessed stress increased, and some significant changes

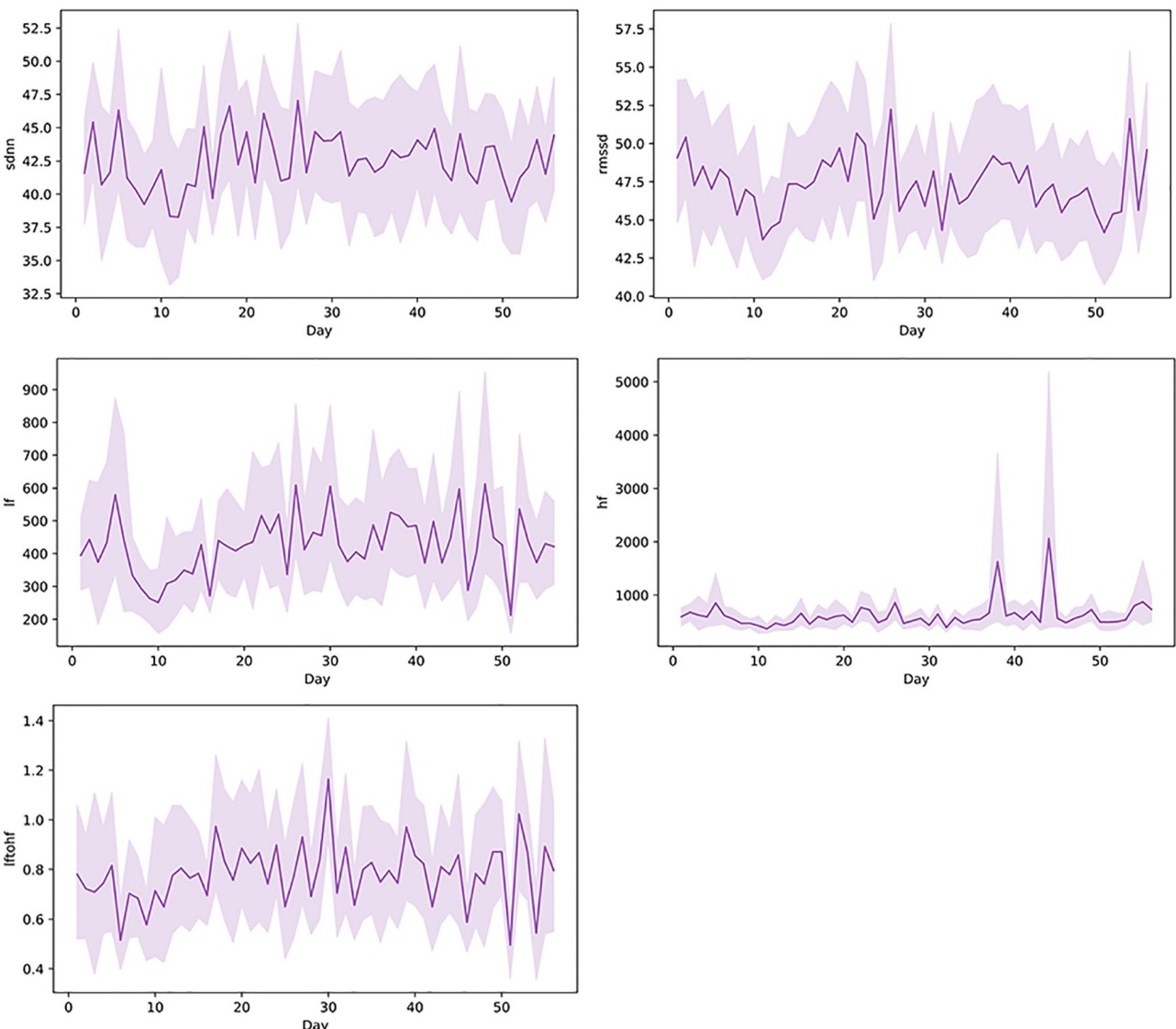

**Fig 2. Trends in HRV parameters.** The mean daily values with 95% confidence intervals of SDNN, RMSSD, LF, HF and LF/HF ratio during the eight-week data collection period (56 days) (n = 28–34).

were detected in HRV. Physical activity decreased when the restrictions were set but also because the pregnancy was proceeding, which is a common finding regarding pregnancy and physical activity. TST decreased as pregnancy proceeded, but the pandemic-related restrictions were not associated with sleep. However, the participants' daily rhythms changed as they started to sleep later as well as wake up later.

Some significant changes occurred in HRV patterns related to the pandemic. The changes in HRV were somehow conflicting, as the increased SDNN might indicate a lower level of stress, whereas increased LF power and an increased LF/HF ratio might indicate higher levels of stress [29, 30]. However, during the follow-up, the SDNN started to decrease, probably because pregnancy proceeded. The clinical significance of these small changes is difficult to evaluate. Further, the function of heart is such a complicated system, and influenced, for example, by individual physiological, mental and hormonal factors, thus interpretation of HRV

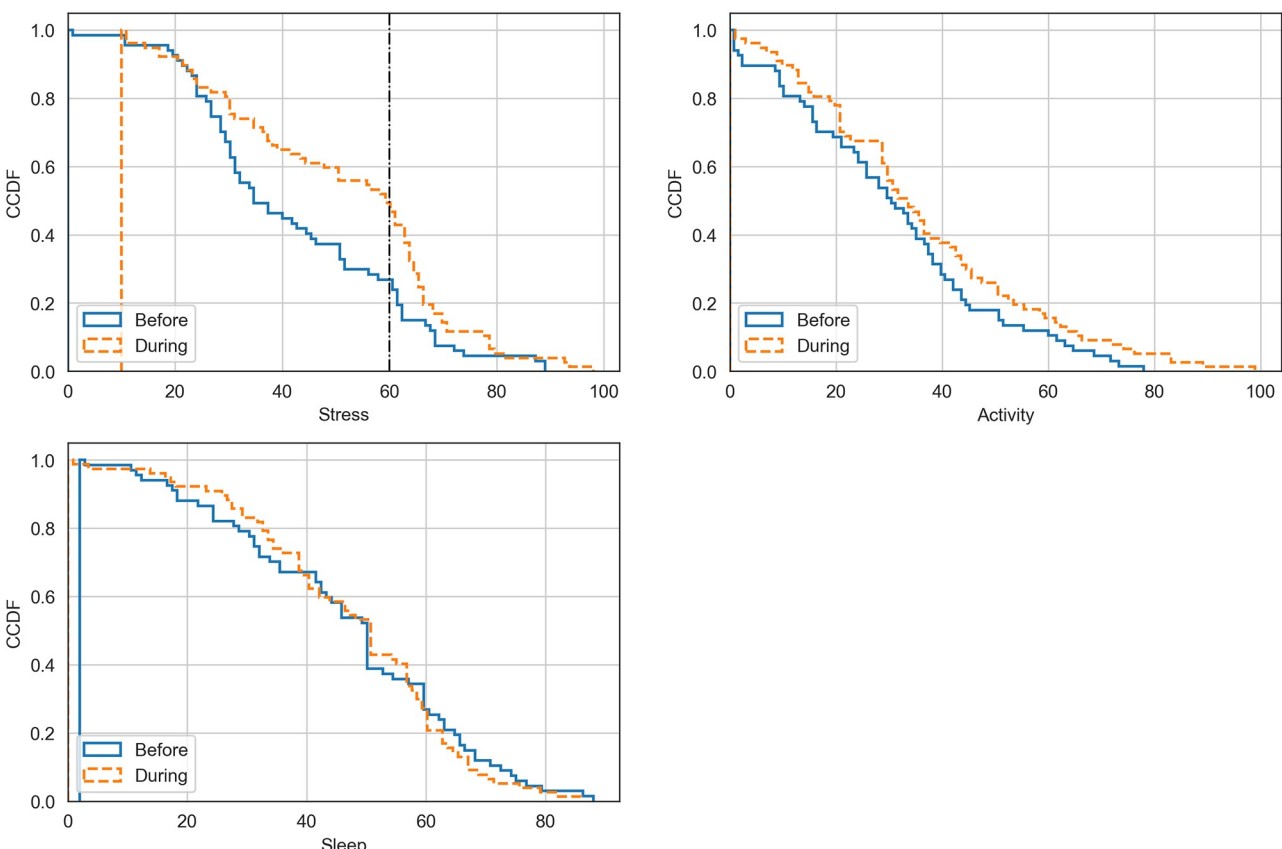

**Fig 3. Subjective evaluations of stress, physical activity and sleep.** Participants' (n = 23) subjectively assessed level of stress, level of physical activity and quality of sleep before and during the pandemic-related restrictions in a scale from 0 to 100 the higher value indicating a higher level of stress, higher level of physical activity and better quality of sleep.

patterns measured in everyday settings is only indicative [22]. The changes detected in this study with pregnant women, however, might be partly explained also by the physiological decrease in HRV as pregnancy proceeds [16]. Both parasympathetic and sympathetic nervous systems contribute to SDNN, thus 24-hour continuous instead of short-term monitoring would provide more accurate values. LF power reflects baroreceptor activity and HF power is more clearly correlated with stress or worry [22].

Pregnant women's increased subjective experience of stress due to pandemic was not reflected as a notable changes in autonomic cardiac control as measured with HRV. Despite of the status of "high-risk" of some pregnant women, the participants in this study generally did not suffer from depressive symptoms or did not seem to have pregnancy-related anxiety, and therefore, they may have sufficient psychological resources for handling special circumstances, including the stay-at-home orders. Furthermore, the women in our study had access to social reserves and resources, such as the statutory public benefit of a maternity grant aimed at offsetting the costs of having a child [31]. This might have played a role in supporting good personal coping amidst the uncertainties and stressful restrictions related to the global pandemic. Women with high-risk pregnancies may especially have higher levels of stress and therefore limited resources for coping with unexpected challenges [32]. Thus, it is important to identify those women who are at risk of inadequately coping with cumulative stressful life events [33].

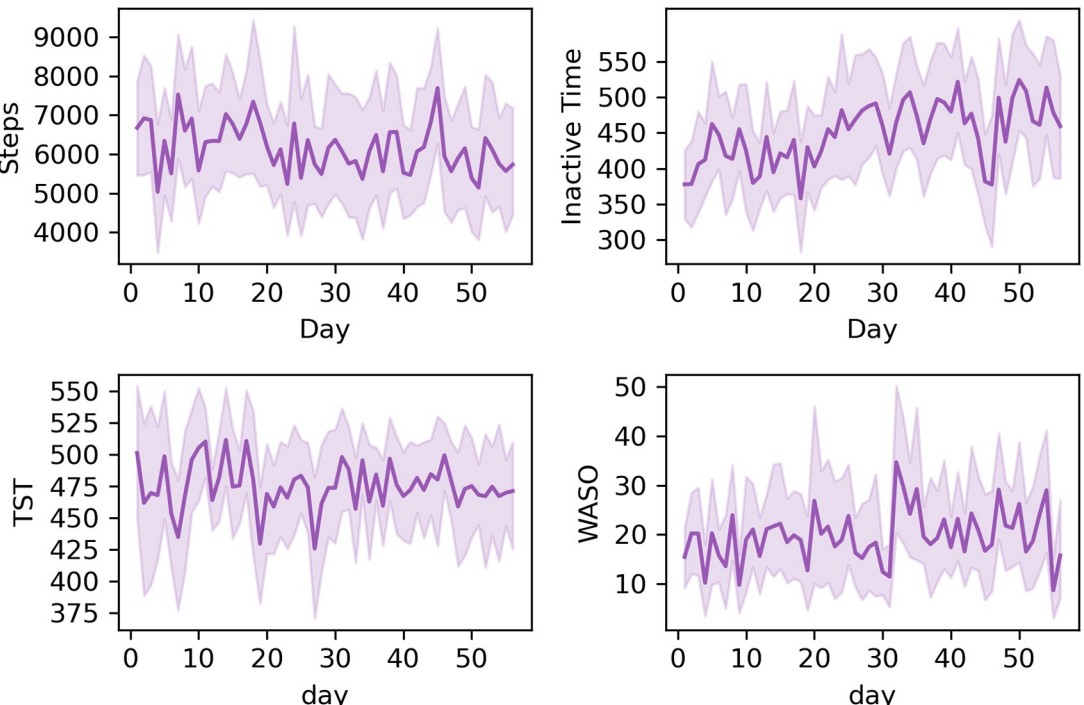

**Fig 4. Trends in physical activity and sleep.** The daily mean with 95% confidence intervals of daily steps, and daily inactive time (n = 28–37), and total sleep time (TST) and wake after sleep onset (WASO) (n = 22–32) during the eight-week data collection period (56 days).

As expected, women's daily step counts decreased and their inactive time increased as pregnancy proceeded [5]. Physical activity decreased significantly when the pandemic-related restrictions were set, but based on the actual step counts, the situation stabilized quite fast. Some of the women might have found new possibilities for exercising at home. Interestingly, women were making changes to their sleep patterns by going to bed a little later and waking up in the morning a little later. Some of the women started their maternity leaves and thus were able to modify their daily rhythms. Some women probably began to work from home according to the pandemic-related regulations, as the women in our study were highly educated. Distant working allowed for flexible working hours and environment—no commute to work, and more flexibility with sleep timing—which allows for the implementation of personalized sleep and activity patterns [34].

In future studies, this IoT platform and device/app pairing could be used for the continuous monitoring and viewing of parameter data by women and clinicians in real time. Understanding and giving pregnant women insight into their own daily patterns of well-being, such as stress, physical activity and sleep during pregnancy, could illuminate the extent to which women are able to cope with stressful events and pregnancy-induced stress/anxiety. Women can also work as partners with clinicians in the personalization of self-care goals during stressful and uncertain times [35].

## Limitations

The sample size was relatively small, and power analysis was not performed. However, due to the continuous measurements by the smartwatch, the amount of data is considerable. The variation in pregnancy weeks between the participants might have caused bias, which limits the

conclusions. Furthermore, increasing pregnancy weeks was a confounding factor, as the pregnancy itself had an impact on a woman's physiological parameters, such as HRV measures [29]. The participating women were generally quite healthy; thus, the findings might differ in other populations. HRV measures are individual and HRV is quite sensitive for many physiological and mental factors, thus not controlling them in the analyses may have caused some bias [17]. With PPG, it is recommended to have a sampling frequency of 25Hz, but with 20Hz, the results could also be deemed reliable [23]. The frequency-domain measures especially might need increased sampling frequency [22]. It has to be noted that the validity of the Samsung Gear Sport has not been confirmed in measuring the heart rate variability, although acceptable validity regarding sleep and step count has been achieved [13, 14]. The generalizability of the results may be limited, but this study provides unique and prospective data about the daily patterns of well-being in pregnant women before and during the COVID-19 pandemic.

## Conclusions

Women in this study were successful in coping with the current disruptions to their lives, possibly due to access to strong supportive social resources. The changes in stress, physical activity and sleep were moderate and in line with increasing pregnancy weeks. The use of IoT technologies for the monitoring of daily patterns of well-being for pregnant women is modern and effective for providing useful parameter information for the promotion of health and wellness.

## Acknowledgments

We would like to acknowledge the voluntary pregnant women who participated in the study and committed to the long data collection period.

## Author Contributions

**Conceptualization:** Hannakaisa Niela-Vilén, Jennifer Auxier, Eeva Ekholm, Pasi Liljeberg, Amir M. Rahmani, Anna Axelin.

**Data curation:** Fatemeh Sarhaddi, Milad Asgari Mehrabadi, Iman Azimi.

**Formal analysis:** Fatemeh Sarhaddi, Milad Asgari Mehrabadi, Aysan Mahmoudzadeh.

**Funding acquisition:** Amir M. Rahmani, Anna Axelin.

**Investigation:** Hannakaisa Niela-Vilén, Fatemeh Sarhaddi, Milad Asgari Mehrabadi, Anna Axelin.

**Methodology:** Hannakaisa Niela-Vilén, Jennifer Auxier, Eeva Ekholm, Fatemeh Sarhaddi, Milad Asgari Mehrabadi, Aysan Mahmoudzadeh, Iman Azimi, Anna Axelin.

**Project administration:** Pasi Liljeberg, Amir M. Rahmani, Anna Axelin.

**Resources:** Pasi Liljeberg, Amir M. Rahmani, Anna Axelin.

**Software:** Fatemeh Sarhaddi, Milad Asgari Mehrabadi, Iman Azimi, Amir M. Rahmani.

**Supervision:** Pasi Liljeberg, Amir M. Rahmani, Anna Axelin.

**Writing – original draft:** Hannakaisa Niela-Vilén, Jennifer Auxier, Fatemeh Sarhaddi, Milad Asgari Mehrabadi.

**Writing – review & editing:** Eeva Ekholm, Aysan Mahmoudzadeh, Iman Azimi, Pasi Liljeberg, Amir M. Rahmani, Anna Axelin.

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
