## [Decision Letter · Decision Letter 0]

25 Nov 2020

PONE-D-20-32152

Pregnant women’s daily patterns of well-being before and during the COVID-19 pandemic in Finland: longitudinal monitoring through smartwatch technology

PLOS ONE

Dear Dr. Niela-Vilen,

Thank you for submitting your manuscript to PLOS ONE. After careful consideration, we feel that it has merit but does not fully meet PLOS ONE’s publication criteria as it currently stands. Therefore, we invite you to submit a revised version of the manuscript that addresses the points raised during the review process.

We look forward to receiving your revised manuscript.

Kind regards,

Kelli K Ryckman

Academic Editor

PLOS ONE

Journal Requirements:

2.Please provide the following information in your Methods section regarding validation of the  weekly question asked related to the COVID-19.

Reviewers' comments:

Reviewer's Responses to Questions

**Comments to the Author**

1. Is the manuscript technically sound, and do the data support the conclusions?

Reviewer #1: Partly

2. Has the statistical analysis been performed appropriately and rigorously? 

Reviewer #1: Yes

3. Have the authors made all data underlying the findings in their manuscript fully available?

Reviewer #1: No

4. Is the manuscript presented in an intelligible fashion and written in standard English?

Reviewer #1: Yes

5. Review Comments to the Author

Reviewer #1: Pregnant women’s daily patterns of well-being before and during the COVID-19

pandemic in Finland: longitudinal monitoring through smartwatch technology

This study examines two cohorts of pregnant women (N=38), whose PPG, sleep and physical activity data were collected over four week periods before and during

national stay-at-home restrictions in response to the COVID-19 pandemic. Overall, the writing is straightforward and a rationale for the study is clear and relevant.

There are several points at which the authors could improve the clarity of the methods as well as their interpretation of the findings. It is unclear for what type of overall study these women were recruited and that could be made more apparent. In terms of the device used to collect the data, it might be helpful to know whether this device has been validated or used for research purposes or to mention earlier than in the limitations section. The authors also include both high and low risk women in this sample and report demographics and mean EPDS and anxiety scores. This has the potential to confound some meaningful data. First, the authors might add validity to their approach by showing statistically that these two groups do not differ in meaningful ways across demographics and clinical characteristics. They might also add these analyses to their outcomes, as there may be distinct differences in the overall stress of women who are high-risk and women who are low-risk in this sample and those differences may be observed in both HRV and sleep. If these groups are statistically no different, than these findings are valid. If not, there is an issue here of examining qualitatively distinct groups that are better off being examined separately. Alternately, the authors can explain their rationale in including all of these women in the same study via providing more context for the overall study (specifics that are aforementioned in this review).

The authors find that SDNN increased across time despite the LF and LF/HF ratio increasing. These results are somewhat counter to one another, but the author discusses this as if there is a linear relationship between stress and HRV (starting in line 340). It might be more helpful to discuss this counterintuitive finding considering what aspects of autonomic functioning contribute to SDNN vs LF HRV ( see Shaffer & Ginsberg, 2017 for a broad but meaningful overview). In fact, HRV may become blunted or hypoactive in the face of acute stressors In this sample, it is possible that there may be other clinical factors at play (increases in depression or anxiety or event symptoms of PTSD) that might impact parameters of HRV in either the time or frequency domains. The authors do not control for any other factors related to HRV, including weight or physical health and it is incredibly difficult to draw conclusions related to HRV without controlling for these factors or holding other factors consistent, including time of day when the data was extracted. The authors might do well to refer to major papers that discuss these recommended methods directly and include these potential confounders in their paper if they have them or address them as limitations if they do not (Laborde, Mosley & Thayer, 2017). While I agree that this article has merit, the oversight in terms of psychophysiological data analyses and interpretation is hard to ignore.

Laborde, S., Mosley, E., & Thayer, J. F. (2017). Heart rate variability and cardiac vagal tone in psychophysiological research–recommendations for experiment planning, data analysis, and data reporting. Frontiers in psychology, 8, 213.

Shaffer, F., & Ginsberg, J. P. (2017). An overview of heart rate variability metrics and norms. Frontiers in public health, 5, 258

6. PLOS authors have the option to publish the peer review history of their article (what does this mean?). If published, this will include your full peer review and any attached files.

Reviewer #1: No

---

## [Author Response · Author response to Decision Letter 0]

18 Dec 2020

Responses to the Editor and the Reviewer

Journal Requirements:

RESPONSE: We have used the given templates and followed the PLOS ONE’s style requirements throughout all the files.

2.Please provide the following information in your Methods section regarding validation of the weekly question asked related to the COVID-19.

 RESPONSE: The weekly questions were developed for this study and therefore not validated. We have added this information in the Methods section.

 RESPONSE: We have revised the cover letter accordingly. 

RESPONSE: We do not have any Supporting information files in our submission.

Review Comments to the Author

Reviewer #1: Pregnant women’s daily patterns of well-being before and during the COVID-19

pandemic in Finland: longitudinal monitoring through smartwatch technology

This study examines two cohorts of pregnant women (N=38), whose PPG, sleep and physical activity data were collected over four week periods before and during

national stay-at-home restrictions in response to the COVID-19 pandemic. Overall, the writing is straightforward and a rationale for the study is clear and relevant.

RESPONSE: We would like to thank the Reviewer for the positive and constructive criticism provided. Our detailed responses to the questions and concerns below. The changes in the manuscript text are made with “track changes”.

There are several points at which the authors could improve the clarity of the methods as well as their interpretation of the findings. It is unclear for what type of overall study these women were recruited and that could be made more apparent.

RESPONSE: We have added more detailed description of the overall study where these women were recruited in the Study design paragraph. 

In terms of the device used to collect the data, it might be helpful to know whether this device has been validated or used for research purposes or to mention earlier than in the limitations section. 

RESPONSE: We have added a sentence about the validity of the smartwatch in the beginning of the Data collection paragraph. In addition, we modified the text regarding validity in the Limitations section.

The authors also include both high and low risk women in this sample and report demographics and mean EPDS and anxiety scores. This has the potential to confound some meaningful data. First, the authors might add validity to their approach by showing statistically that these two groups do not differ in meaningful ways across demographics and clinical characteristics. They might also add these analyses to their outcomes, as there may be distinct differences in the overall stress of women who are high-risk and women who are low-risk in this sample and those differences may be observed in both HRV and sleep. If these groups are statistically no different, than these findings are valid. If not, there is an issue here of examining qualitatively distinct groups that are better off being examined separately. Alternately, the authors can explain their rationale in including all of these women in the same study via providing more context for the overall study (specifics that are aforementioned in this review).

RESPONSE: We have added the background characteristics of both high-risk and low-risk groups separately in the Table 1. There were no differences between the groups except the gestational weeks during the study period since the recruitment of the participants in the high-risk group was initiated and completed earlier. However, we used the group also as an independent variable in the statistical models to confirm there were no differences between these groups.

The authors find that SDNN increased across time despite the LF and LF/HF ratio increasing. These results are somewhat counter to one another, but the author discusses this as if there is a linear relationship between stress and HRV (starting in line 340). It might be more helpful to discuss this counterintuitive finding considering what aspects of autonomic functioning contribute to SDNN vs LF HRV ( see Shaffer & Ginsberg, 2017 for a broad but meaningful overview). In fact, HRV may become blunted or hypoactive in the face of acute stressors In this sample, it is possible that there may be other clinical factors at play (increases in depression or anxiety or event symptoms of PTSD) that might impact parameters of HRV in either the time or frequency domains. The authors do not control for any other factors related to HRV, including weight or physical health and it is incredibly difficult to draw conclusions related to HRV without controlling for these factors or holding other factors consistent, including time of day when the data was extracted. 

The authors might do well to refer to major papers that discuss these recommended methods directly and include these potential confounders in their paper if they have them or address them as limitations if they do not (Laborde, Mosley & Thayer, 2017). While I agree that this article has merit, the oversight in terms of psychophysiological data analyses and interpretation is hard to ignore.

Laborde, S., Mosley, E., & Thayer, J. F. (2017). Heart rate variability and cardiac vagal tone in psychophysiological research–recommendations for experiment planning, data analysis, and data reporting. Frontiers in psychology, 8, 213.

Shaffer, F., & Ginsberg, J. P. (2017). An overview of heart rate variability metrics and norms. Frontiers in public health, 5, 258

RESPONSE: Thank you for pointing this out. We have now modified the Discussion section in order to emphasize the challenges in interpreting the HRV results. We were not able to control all the confounding variables, however, we used only night-time data thus the day of the time was controlled. The Limitations section has also been modified to provide a clear picture of which parameters were controlled and which not. Further, we have added the papers the Reviewer suggested in the discussion and found them very relevant and useful for our manuscript. 

---

## [Decision Letter · Decision Letter 1]

20 Jan 2021

Pregnant women’s daily patterns of well-being before and during the COVID-19 pandemic in Finland: longitudinal monitoring through smartwatch technology

PONE-D-20-32152R1

Dear Dr. Niela-Vilen,

We’re pleased to inform you that your manuscript has been judged scientifically suitable for publication and will be formally accepted for publication once it meets all outstanding technical requirements.

Kind regards,

Kelli K Ryckman

Academic Editor

PLOS ONE

Additional Editor Comments (optional):

Reviewers' comments:

Reviewer's Responses to Questions

**Comments to the Author**

1. If the authors have adequately addressed your comments raised in a previous round of review and you feel that this manuscript is now acceptable for publication, you may indicate that here to bypass the “Comments to the Author” section, enter your conflict of interest statement in the “Confidential to Editor” section, and submit your "Accept" recommendation.

Reviewer #1: All comments have been addressed

2. Is the manuscript technically sound, and do the data support the conclusions?

Reviewer #1: Yes

3. Has the statistical analysis been performed appropriately and rigorously? 

Reviewer #1: Yes

4. Have the authors made all data underlying the findings in their manuscript fully available?

Reviewer #1: No

5. Is the manuscript presented in an intelligible fashion and written in standard English?

Reviewer #1: Yes

6. Review Comments to the Author

Reviewer #1: Point by point, the authors have appropriately addressed comments offered and integrated this reviewer's suggestions appropriately. I think this paper makes a contribution to the field in regards to the limited knowledge that we have about psychophysiology during the perinatal period.

7. PLOS authors have the option to publish the peer review history of their article (what does this mean?). If published, this will include your full peer review and any attached files.

Reviewer #1: No

---

## [Editor Report · Acceptance letter]

25 Jan 2021

PONE-D-20-32152R1 

Pregnant women’s daily patterns of well-being before and during the COVID-19 pandemic in Finland: longitudinal monitoring through smartwatch technology 

Dear Dr. Niela-Vilen:

I'm pleased to inform you that your manuscript has been deemed suitable for publication in PLOS ONE. Congratulations! Your manuscript is now with our production department. 

Kind regards, 

on behalf of

Dr. Kelli K Ryckman 

Academic Editor

PLOS ONE